# Allergic Proctocolitis: Literature Review and Proposal of a Diagnostic–Therapeutic Algorithm

**DOI:** 10.3390/life13091824

**Published:** 2023-08-29

**Authors:** Simona Barni, Francesca Mori, Mattia Giovannini, Lucia Liotti, Carla Mastrorilli, Luca Pecoraro, Francesca Saretta, Riccardo Castagnoli, Stefania Arasi, Lucia Caminiti, Mariannita Gelsomino, Angela Klain, Michele Miraglia del Giudice, Elio Novembre

**Affiliations:** 1Allergy Unit, Meyer Children’s Hospital IRCCS, 50139 Florence, Italy; simonabarni@hotmail.com (S.B.); francesca.mori@unifi.it (F.M.); mattiag88@hotmail.it (M.G.); elio.novembre@unifi.it (E.N.); 2Department of Health Sciences, University of Florence, 50139 Florence, Italy; 3Pediatric Unit, Department of Mother and Child Health, Salesi Children’s Hospital, 60123 Ancona, Italy; lucialiotti@libero.it; 4Pediatric and Emergency Department, Pediatric Hospital Giovanni XXIII, AOU Policlinic of Bari, 70126 Bari, Italy; carla.mastrorilli@gmail.com; 5Pediatric Unit, Department of Surgical Sciences, Dentistry, Gynecology and Pediatrics, University of Verona, 37126 Verona, Italy; luca.pecoraro@aovr.veneto.it; 6Pediatric Department, Latisana-Palmanova Hospital, Azienda Sanitaria Universitaria Friuli Centrale, 33100 Udine, Italy; francescasaretta@gmail.com; 7Department of Clinical, Surgical, Diagnostic and Pediatric Sciences, University of Pavia, 27100 Pavia, Italy; riccardo.castagnoli@yahoo.it; 8Pediatric Clinic, Fondazione IRCCS Policlinico San Matteo, 27100 Pavia, Italy; 9Translational Research in Pediatric Specialties Area, Division of Allergy, Bambino Gesù Children’s Hospital, IRCCS, 00165 Rome, Italy; stefania.arasi@opbg.net; 10Allergy Unit, Department of Pediatrics, AOU Policlinico Gaetano Martino, 98124 Messina, Italy; lucycaminiti@yahoo.it; 11Department of Life Sciences and Public Health, Pediatric Allergy Unit, University Foundation Policlinico Gemelli IRCCS, Catholic University of the Sacred Heart, 00168 Rome, Italy; 12Department of Woman, Child and General and Specialized Surgery, University of Campania Luigi Vanvitelli, 80138 Naples, Italy; klainangela95@gmail.com (A.K.); michele.miragliadelgiudice@unicampania.it (M.M.d.G.)

**Keywords:** allergic proctocolitis, rectal bleeding, food allergy, non-lgE food allergy, elimination diet

## Abstract

Allergic proctocolitis (AP) is a benign condition, frequent in childhood, that is classified as a non-IgE-mediated food allergy. The prevalence is unknown; however, its frequency appears to be increasing, especially in exclusively breastfed infants. Clinical manifestations typically begin in the first few months of life with the appearance of bright red blood (hematochezia), with or without mucus, in the stool of apparently healthy, thriving infants. Most cases of AP are caused by cow’s milk proteins; however, other allergens, such as soy, egg, corn, and wheat, may be potential triggers. Diagnosis is based on the patient’s clinical history and on the resolution of signs and symptoms with the elimination of the suspected food antigen from the diet and their reappearance when the food is reintroduced into the diet. The treatment of AP is based on an elimination diet of the trigger food, with resolution of the symptoms within 72–96 h from the beginning of the diet. The prognosis of AP is good; it is a self-limiting condition, because most children can tolerate the trigger food within one year of life, with an excellent long-term prognosis. The purpose of this review is to provide an update on the current knowledge and recommendations in epidemiological, diagnostic, and therapeutic terms to the pediatricians, allergists, and gastroenterologists who may find themselves managing a patient with AP.

## 1. Introduction

Allergic proctocolitis (AP), formerly known as allergic colitis, allergic eosinophilic proctitis/proctocolitis, or food protein-induced proctitis/proctocolitis [1,2], was first described by Rubin [3] in 1940 and later by Gryboski in 1966 and 1967 [4,5].

## 2. Epidemiology

The real prevalence and incidence of AP is not known [2], although it is among the most frequent causes of rectal bleeding in children.

The epidemiology of AP across studies in the literature varies widely, probably due to differences in case identification methodology (e.g., whether only AP caused by cow’s milk or different food allergens is considered) and in the diagnosis of AP (whether based only on the elimination diet or on a histological criterion, or on an elimination diet and subsequent reintroduction into the diet).

In a prospective population-based study (in which 13,019 newborns were enrolled over a two-year period), the prevalence of AP due to cow’s milk protein was found to be 1.6 per 1000 infants [6].

A US prospective cohort study of children with rectal bleeding demonstrated that 64% had AP, a diagnosis based on the histologic biopsy report obtained via flexible sigmoidoscopy [7], whereas only 18% of children with hematochezia had AP, a diagnosis based on a diet of elimination and subsequent reintroduction into the diet [8].

A Turkish retrospective study found an AP frequency of 0.18% over a 10-year period [9], a diagnosis based on symptom resolution with the elimination of the trigger food from the child’s diet and subsequent reappearance at the time of its reintroduction.

A prospective observational cohort study of healthy children reported a cumulative incidence of 17% over a three-year period. The diagnosis of AP was made by the family pediatrician for the finding of occult blood in the stools with no other explicable cause, which was resolved with an elimination diet removing cow’s milk. This cumulative incidence decreased to 7% if the presence of macroscopically visible blood in the stool was used as a diagnostic criterion [10]. The authors explained the high incidence rate in their not having performed the diagnostic oral food challenge (OFC), with the consequent risk of overestimating AP.

## 3. Pathogenesis

The exact immunological mechanism responsible for this condition has not yet been elucidated. AP is usually classified as a non-IgE-mediated food allergy [11,12]. It is believed that the main risk factors for the development of the disease are an immaturity of the innate and adaptive immune systems, impaired intestinal permeability, and a genetic predisposition, in combination with a sensitivity, to food antigens [13].

AP is caused by an immunological lack of tolerance to food antigens, which seems to play a fundamental role in the development of AP [14]. Several studies have shown that different cells of the immune system are involved in the induction of oral tolerance to foods [14]. Research on pediatric patients with multiple food allergies has shown that the main immunological abnormality in the small intestine is an abnormal production of transforming growth factor-beta (TGF-β) by regulatory cells [15]. It has therefore been hypothesized that this lack of production in relation to an insufficient innate immune response against the intestinal commensal microbial flora could contribute to an altered development of oral tolerance to foods [16]. Reduction of TGF-β1 expression was also seen in mononuclear cells and the epithelium in food allergies by immunohistochemistry and in situ hybridization. The dominant mucosal abnormality in food allergic children was, thus, not Th2 deviation but impaired generation of Th3 cells [15].

A study of pediatric patients with multiple food allergies showed, by flow cytometry, immunohistochemistry, and in situ hybridization, that TGF-β expression is significantly reduced in the duodenal mucosa, both in immediate and late adverse food reactions [17]. The study conducted by Ozen et al. [18] confirms that the reduced expression of TGF-β may be involved in the sensitization to food proteins.

It has been shown that tumor necrosis factor-alpha (TNF-α), a key cytokine in the pathogenesis of inflammatory bowel disease, alters the tight junctions between epithelial cells [19]. Due to this property, TNF-α therefore appears to be involved in the pathogenesis of AP through the alteration of the intestinal epithelial barrier capacity. In fact, the increased levels of TNF-α lead to increased production and activation of other proinflammatory cytokines and factors that promote intestinal inflammation [19]. In addition to activating the endogenous inflammatory cascade, a potential proinflammatory action of TNF-α includes the alteration of the intestinal epithelial TJ barrier. TNF-α has been shown to produce an increase in epithelial TJ permeability in various cell types, including in intestinal epithelial cells [19]. In accordance with this evidence, a study [20] examined the concentration of TNF-α in three groups of patients: a control group, patients with IgE-mediated cow’s milk allergy, and patients with gastrointestinal food allergy (including AP). The study measured the concentration of TNF-α in peripheral blood mononuclear cells and found that patients with gastrointestinal food allergy had a significantly higher concentration of TNF-α than that found in the other patient groups, while the concentrations of other cytokines, such as interferon-gamma (IFN-γ) and interleukin 17 (IL-17), were not found to be statistically different in the different groups of patients. Interleukin 3 (IL-3), interleukin 5 (IL-5), and interleukin 13 (IL-13) were cytokines produced in a statistically significant way by peripheral blood mononuclear cells stimulated in vitro with cow’s milk in patients with food allergies.

Gut microbiota has also been studied as a risk factor involved in the pathogenesis of AP development.

In one study [21], the intestinal microbiota and secretory IgA present in the stools of 15 exclusively breastfed infants with hematochezia were compared with 15 exclusively breastfed infants without hematochezia. All patients were born vaginally. It was observed that *Bacterioides fragilis* was more highly represented in the stools of patients without hematochezia than in those with hematochezia (*p* < 0.05). In the healthy control group, the predominant species belonging to the Enterobacteriaceae group was *Escherichia coli*, while in patients with hematochezia, it was Klebsiella (*p* < 0.05). The concentration of secretory IgA was high in one third of healthy patients. In conclusion, the pathogenetic mechanism of rectal bleeding in exclusively breastfed infants may be associated with different compositions of the intestinal flora.

The role of hypoallergenic formulas supplemented with probiotic strains has recently been studied; in particular, Baldassarre et al. [22] studied *Lactobacillus rhamnosus* GG (LGG) added to an extensively hydrolyzed casein formula (EHCF) in patients with hematochezia compared to a hydrolyzed casein formula not added to LGG. The use of EHCF + LGG resulted in a significant improvement in hematochezia compared to the use of EHCF alone.

Furthermore, a study [23] examined the effect of fecal microbiota transplantation (FMT) from healthy patients to patients with PA. After treatment with FMT, the clinical manifestations in 17 AP patients improved within two days, and no recurrence was noted in the following 15 months. In the stools of AP patients, after FMT, Proteobacteria decreased, while Firmicutes increased (representing the major percentage of the gut microbiota). At the genus level, *Bacterioides, Escherichia* and *Lactobacillus* increased in the stool of AP patients after FMT, while *Clostridium*, *Veillonella*, *Streptococcus*, and *Klebsiella* decreased dramatically.

## 4. Clinical Manifestations

The main clinical manifestation of AP is the presence of bright red blood (hematochezia), with or without mucus, mixed in with stool, with or without diarrhea, in apparently healthy, regularly growing infants [11,12,24,25]. A minority of patients may also present with refusal to eat and irritability, abdominal pain, pain on defecation, and flatulence [2,26]. Warning signs and symptoms suggesting a diagnosis other than AP are poor general clinical conditions, severe diarrhea, anemia, repeated vomiting, abdominal distension, perianal disease, weight loss, or failure to thrive [2,27].

Usually, clinical manifestation onset is within the first six months of life (most patients present symptoms between the first and fourth week of life [26]; AP affects both exclusively breastfed and formula-fed infants, although it appears to affect exclusively breastfed infants slightly more (56.8–60%) [9,26].

Although AP occurs in most cases in infants in the first months of life, it can occur in older children, as demonstrated by Ravelli et al. [28], in which AP was diagnosed in 18% of patients aged 2 to 14 years with rectal bleeding.

Among the atopic comorbidities, eczema is present in 22% to 52% of patients with AP [9,10,26]. An atopic family history is also present in 25% to 50% [9,10,26] of patients.

In breastfed infants, cow’s milk is the food antigen most commonly associated with AP, but other foods, such as soy, egg, corn, and wheat, may also be implicated in the development of clinical manifestations [6,29,30,31,32,33]. In addition, 5% to 42% of patients have more than one culprit food antigen [29,30,31,32,33,34,35]. Risk factors for the development of AP to multiple food allergens have been found to be the concomitant presence of atopic dermatitis, high levels of eosinophils at diagnosis, and allergic sensitization (prick test or specific IgE) to the culprit food [29,33].

In bottle-fed infants, AP is caused by cow’s milk proteins and soy; extensively hydrolyzed formulas cause AP in 5–10% of cases [26,36].

The summary of the main clinical features is reported in Table 1 (modified from [1,37]).

## 5. Diagnosis

The diagnosis of AP is almost always clinical, based on the presence of typical symptoms that resolve with the elimination of the trigger antigen from the diet. Although the OFC is not generally performed for the diagnosis of AP, early reintroduction of the food at home, following a 2–4-week period of an elimination diet, has been proposed to minimize the risk of overdiagnosis [8,38,39].

Although there are no validated diagnostic criteria in the literature, some criteria are commonly used in clinical practice to support the diagnosis of AP, including those proposed by the position paper of the European Academy of Allergy and Clinical Immunology (EAACI) [12], which are as follows:mild rectal bleeding in an otherwise healthy patient;resolution of clinical manifestations after elimination of trigger food(s) (if breastfed, resolution after maternal elimination diet);reappearance of symptoms on reintroduction of the trigger food(s);exclusion of other causes of rectal bleeding.

A diagnostic confirmation may not be necessary in some milder cases of AP, in which the reintroduction, after an adequate period of an elimination diet, has the sole purpose of verifying any tolerance.

## 6. Non-Invasive Tests

Laboratory tests: Blood tests can support the diagnosis of non-IgE-mediated allergy, although the results are not pathognomonic. The blood count can detect anemia in the case of chronic bleeding and peripheral hypereosinophilia [35,40,41]. Furthermore, in some cases, it is possible to find hypoalbuminemia with hyperproteinemia and an increase in total IgE [11,26,42]. The blood count may be useful in the case of persistent hematochezia for at least one month, as suggested by Miceli Sopo et al. [38]; in the case of anemia, iron supplementation will be carried out.

Allergy tests: Allergy tests, including the skin prick test (SPT) and specific serum IgE, are usually negative and therefore are not recommended in patients suspected of having AP at the time of diagnosis [12]. However, allergic sensitization in AP varies from 10 to 35%, depending on the case series [9,33,41]. Two studies [34,35] have demonstrated how sensitization to trigger food can be used as a prognostic factor; in fact, patients with a positive SPT exhibit a later acquisition of tolerance compared with non-sensitized patients. Furthermore, an EAACI position paper states that allergy testing may be considered in breastfed patients with signs and symptoms associated with an IgE-mediated allergy, in the presence of comorbidities such as atopic dermatitis, and after a long period of elimination of the food before its reintroduction, as already suggested by Miceli Sopo et al. [38] and Nowak [26]. The utility of the patch test is controversial because there are no studies that have validated this test; therefore, it is not recommended to perform it as a routine test for the diagnosis of AP [11].

Stool Tests: Stool analysis can show the presence of polymorphonuclear leukocytes, typically eosinophils, in patients with AP [43].

Fecal calprotectin (FC) is a calcium- and zinc-binding protein that accounts for 60% of neutrophil cytosolic proteins and is a marker of intestinal inflammation [44]. FC may be elevated in patients with AP, but its clinical utility has not been established. A study carried out on 32 children affected by AP from cow’s milk demonstrated that FC was statistically higher in the group of patients with AP than in the control group and that this was statistically significantly reduced in patients with AP after 4 weeks from the beginning of the elimination diet [45]. However, the utility of this test for diagnosing AP is limited due to the overlap of FC results in AP patients and controls, in part because FC levels are generally higher in infants younger than 6 months than in healthy older children [46]. FC, being a marker of intestinal inflammation, was found to be elevated in preterm infants with intestinal bleeding, even when blood was present in trace amounts [47,48,49,50,51].

Abdominal ultrasound with color Doppler ultrasound: Abdominal ultrasound with color Doppler ultrasound is used to evaluate intestinal inflammation that causes a thickening of the intestinal mucosa visible on ultrasound [52]. A retrospective analysis of 13 children with AP showed that 92.3% had ultrasound abnormalities such as increased vascularization and thickening of the intestinal walls, especially in the descending colon and sigmoid colon. Colonoscopy with biopsy was performed in these 13 children and showed alterations compatible with AP [53]. All 13 children were then placed on an elimination diet, and 7 of 13 had repeat abdominal ultrasound with color Doppler ultrasound that showed changes in the vascularity and thickness of the intestinal mucosa. Because the pathophysiology of AP is generally related to intestinal inflammation, ultrasonography can confirm this inflammation, which, when associated with the clinic, can suggest the diagnosis of AP [37,52]. Despite this, the alterations visible on abdominal ultrasound with color Doppler ultrasound are not specific to AP, as they can be present in all cases of intestinal inflammation, such as infectious colitis [37,52]. Therefore, more studies are needed to support the routine use of ultrasound in the diagnosis of AP. Furthermore, an instrumental experience dedicated to the pediatric patient by the radiologist is necessary for this type of diagnostic examination.

## 7. Invasive Tests

Colonoscopy or flexible sigmoidoscopy with biopsy is useful in patients with atypical signs and symptoms, such as constipation or diarrhea with bloodless mucous stools or in the case of severe intestinal bleeding, anemia despite an elimination diet [7], or if after 72–96 h there is no clinical response to the maternal elimination diet in breastfed infants or 72–96 h after amino acid formula introduction in bottle-fed infants [2]. Colonoscopy/flexible sigmoidoscopy performed in patients with AP shows a picture of colitis with erythema and mucosal edema with erosions, ulcerations, and loss of vascularization [8,30,54]. These endoscopic changes are usually confined to the distal colon, although they may also extend proximally. Biopsy typically reveals an eosinophilic infiltrate, including eosinophilic abscesses, in the lamina propria and muscularis mucosae, and lymph node hyperplasia [6,28,30,31,55]. Dergent et al. [56] described a case of AP showing, in addition to lymph node hyperplasia and eosinophilic infiltrate in the lamina propria, the presence of granulomatous infiltrate, histiocytes, and multinucleated giant cells in the submucosa. Although histopathologic features are not pathognomonic of PA, Odze et al. [40] suggest that finding more than 60 eosinophils on 10 high-power fields is sufficient to diagnose AP in 97.4% of cases.

The main characteristics of invasive and non-invasive tests for the diagnosis of AP are reported in Table 1 (modified from [1,37]).

## 8. Differential Diagnosis

A differential diagnosis should be made with the following conditions [27,57,58,59,60].

Anal fissures represent the most common cause of bleeding in children younger than one year of age. The diagnosis is clinical by examining the perianal mucosa. Treatment is with the use of stool softeners, because it is often associated with constipation.

Intussusception: Patients with intussusception typically present with episodes of sudden cramping or severe and progressive abdominal pain accompanied by inconsolable crying, with flexion of the legs over the abdomen. The stools typically look like currant jelly. It usually affects children between 6 and 26 months, and it is infrequent before 3 months of life.

Infectious enteritis: Several pathogens can cause gastrointestinal bleeding, including typical enteric pathogenic bacteria (e.g., *Campylobacter jejuni*, *Clostridium difficile*, *Escherichia coli*, *Helicobacter pylori*, *Salmonella*, *Staphylococcus aureus*, *Yersinia enterocolitica*) and occasionally rotavirus. Infections should be suspected when the child has rectal bleeding accompanied by fever, abdominal pain, and tenesmus, especially if there is a history of contact with people who have the same symptoms.

Meckel’s diverticulum: This presents as painless rectal bleeding in healthy children. Presentation before 6 months of age is rare. It can also manifest in rare cases with perforation, intussusception on the diverticulum itself, or as significant indolent intestinal bleeding.

Food protein-induced enterocolitis syndrome (FPIES): This can present in acute and chronic forms. The acute form is characterized by vomiting, diarrhea (with or without blood), hypothermia, lethargy to dehydration, and hypovolemic shock. The chronic form is characterized by diarrhea (with or without blood), occasional vomiting, and weight loss with poor growth. Patients with FPIES appear to suffer more than patients with AP.

Food protein-induced enteropathy (FPE): This is characterized by malabsorption, intermittent vomiting, poor growth and, rarely, bloody stools. The diagnosis is clinical. Endoscopy with a biopsy of the proximal small intestine confirms the villous change.

Eosinophilic gastroenteritis: This is a chronic gastrointestinal immunological condition characterized by eosinophilic infiltration at the level of the mucous, muscular, or serous layer of the gastrointestinal tract. Inflammation can cause vomiting, abdominal pain, diarrhea, intestinal bleeding (hematemesis or hematochezia), anemia, hypoalbuminemia, ascites, or poor growth.

Necrotizing enterocolitis: This affects infants born prematurely in 90% of cases. It occurs in 75% of cases within the first month of life.

Chronic inflammatory bowel disease with early onset: This is a rare cause of rectal bleeding in children characterized by diarrhea (with or without blood), poor growth, and perianal abnormalities (e.g., anorectal fistula or abscesses). It can be associated with autoimmune diseases (e.g., type I diabetes mellitus, sclerosing cholangitis, thyroiditis, arthritis). It occurs in children younger than 2 years of age.

Henoch-Schönlein Purpura: This is a form of vasculitis of the small blood vessels of the skin, bowels, joints, and kidneys. It is an uncommon cause of rectal bleeding in children but can cause life-threatening bleeding, and its treatment is specific. The mean age of onset is 6 years.

Intestinal parasitosis: There are several types of intestinal parasitosis that can cause bloody stools, including Entamoeba Histolytica. Although it is seen very rarely in developed countries, intestinal parasitosis is a very important cause of intestinal bleeding in the Middle East and Asia.

Other causes of rectal bleeding include the ingestion of maternal blood during lactation through nipple fissures, intestinal duplication cysts, vascular malformations, and lymph node hyperplasia.

## 9. Treatment

The recommended treatment of AP generally consists of removing the trigger food from the diet [12].

The approach differs depending on whether the infant is exclusively breastfed or formula-fed [27] (see Figure 1).

### 9.1. Exclusively Breastfed Infants

Breastfeeding should be encouraged if the mother is willing to eliminate the trigger food from her diet. Cow’s milk should be eliminated from the maternal diet first, as well as all dairy and baked products containing milk. Milk from other mammals (e.g., goat, sheep, camel) should also be eliminated due to cross-reactivity with cow’s milk. For infants with severe signs and symptoms, it may be helpful to advise using an amino acid formula rather than breast milk for 3–5 days (time required for the antigen to be cleared from the breast milk), and the mother can pump to maintain her milk ejection reflex. With complete elimination of the trigger food antigen from the maternal diet, hematochezia resolves within 72–96 h, whereas fecal occult blood takes many weeks to clear [1,11,26]; its execution can be confounding and is therefore not indicated [27]. The last sign to disappear is mucus, which takes 30 days, according to the study conducted by Uncuoglu et al. [29].

Most breastfed infants respond to the elimination of cow’s milk from the mother’s diet; only a few cases require the elimination of other food antigens [29,30,35]. A study that deviates from these data is that of Martin et al. [10], in which only 47% of patients resolved the clinical manifestations with the cow’s milk elimination diet alone, 40% resolved with the elimination of milk and soy, and 13% resolved with the elimination of milk, egg, and soy. In other studies, it was necessary to eliminate other foods such as corn, tree nuts, and fish [34]. The different frequencies of trigger foods may depend on the different habits found in the geographical areas where the studies are conducted, as happens for other non-IgE-mediated food allergies (e.g., FPIES) [61].

If the signs and symptoms do not resolve, the first step is to check the mother’s adherence to the elimination diet. If the mother is adherent to the diet and has been eliminating the allergen for at least 2 weeks, but the clinical manifestations persist, then soy, followed by egg or other foods, should be eliminated from the diet [62,63]. Some children may continue to have signs and symptoms despite maternal adherence to the elimination diet. For this group of patients, the approach has not been defined and should be decided on a case-by-case basis after discussing the various options with the parents. The different possibilities are as follows:

(a) Switch from breastfeeding to an extensively hydrolyzed formula or amino acid formula. This option should be considered if the mother finds the elimination diet too burdensome or if she is considering stopping breastfeeding for other personal reasons (e.g., going back to work);

(b) Continue to breastfeed despite the persistence of clinical manifestations. This option is controversial but may be appropriate for children with mild forms, with the agreement of the parents, as it may be psychologically taxing. In support of this recommendation is the study by Arvola et al. [8], which examined 40 children with rectal bleeding, 68% of whom were exclusively breastfed. At enrollment, children were randomized to receive a cow’s milk elimination diet (19 patients) or to continue the unchanged diet (21 patients) for one month. Follow-up was performed after one month and one year. The cow’s milk elimination diet showed no effect on the duration and severity of bleeding at follow-up. The duration of hematochezia in the two patient groups was similar (5.6 days for patients with the elimination diet versus 5.5 in free-diet patients, *p* = 0.94), and the number of bloody stools was not statistically different in the two groups. Only one patient who continued the free diet required iron supplementation for the development of anemia; therefore, the risk of developing anemia was also very low. Thus, the challenge is essential for children, even before one year of age, who become asymptomatic during the elimination diet period, to reduce the number of misdiagnoses and unnecessary diets.

Another prospective population study [6] demonstrated that the hemoglobin levels measured at one year of age in children who continued the cow’s milk elimination diet and those who introduced milk into their diet earlier did not differ (12.26 mg/dL in the elimination diet group versus 12.25 mg/dL in the non-elimination diet group, *p* = 0.98).

### 9.2. Infants Fed with Formula or Mixed Feeding

Formula milk should be replaced with an extensively hydrolyzed formula. Using soy-based formulas is generally not recommended, because a significant proportion of patients (15%) who are intolerant of cow’s milk also do not tolerate soy [64]. Older studies report an even higher percentage, up to 40% [30,36,65]. Approximately 5–10% of patients [26,36] do not respond to the extensively hydrolyzed formula and, therefore, must switch to the amino acid formula. Limited evidence suggests that supplementation with probiotics (e.g., LGG) may promote the acquisition of tolerance [22,66]. In the case of no improvement of signs and symptoms or an unsure diagnosis, it is advisable to refer the patient to a gastroenterologist specialist. Also in this case, another possibility may be to continue with formula milk or mixed feeding.

A summary of a possible dietary approach in AP therapy is shown in Table 2 (modified from [37]).

In 2018, an algorithm for the treatment of AP, based on the duration of hematochezia, was proposed by Miceli Sopo et al. [38]. In the case of a duration less than or equal to one month, the authors suggest waiting for spontaneous resolution, without making changes in the diet; if the hematochezia exceeds one month, the authors suggest the elimination diet, and if the clinical manifestations disappear, the OFC. If the clinical manifestations reappear after the latter, it is suggested to go back to the elimination diet for 3 months. This approach is suggested since, in one study [6], only 21.5% of children had signs and symptoms on reintroduction, which took place after about 3 months. Furthermore, it is also proposed to carry out the OFC at home if the specific IgE for the trigger food is negative, in line with what was reported by Nowak et al. [26].

In 2020, Mennini et al. [2] proposed a diagnostic algorithm, according to which, if the patient with AP presents clinical manifestations lasting longer than 4 weeks, in accordance with Miceli Sopo et al. [38], the maternal elimination diet of cow’s milk in breastfed patients or use of an extensively hydrolyzed formula in formula-fed patients is recommended. If the dietary intervention is effective, and the clinical manifestations disappear in 72–96 h, the authors recommend performing the OFC. If they reappear, it is suggested to go back to the elimination diet for 3 months before performing the OFC again. If, on the other hand, the clinical manifestations persist, and the child is breastfed, it is suggested to also exclude soy and eggs; otherwise, if the infant is fed with extensively hydrolyzed formula, one should switch to an amino acid formula. When the milk, soy, and egg elimination diet or the amino acid diet are ineffective, a colonoscopy is recommended to rule out other diagnoses. The authors do not specify the place where the OFC must be performed nor whether it is necessary to demonstrate the negativity of the specific allergy tests.

Figure 1 shows a possible algorithm for the diagnosis and treatment of AP.

## 10. Reintroduction

The reintroduction of the trigger food must be carried out using the timing, setting, and methods discussed below, also considering which approach to use in the event of a recurrence [27] (see Figure 1).

### 10.1. Timing

For children who clinically respond to the elimination diet with cow’s milk or other food antigens, the guidelines suggest the reintroduction of the trigger food for diagnostic confirmation. However, various studies indicate that the traditional empirical approach of pediatricians is to insert the trigger food directly around one year of life [27,31]. In some situations, reintroduction of small amounts of the trigger food occurs before one year of life and, occasionally, before the sixth month of life [30], particularly if the food is inadvertently introduced into the diet without recurrence of symptoms.

### 10.2. Duration of the Elimination Diet

It is common practice to continue the elimination diet, even in breastfed infants, up to one year of age [7,30,67]. In the study by Kaya et al. [31], out of 60 patients with AP diagnosed with the OFC with cow’s milk, only 53% achieved tolerance at 1.25 years, 25% achieved it at 2 years, 5% at 3 years, and 1.7% at 4 years. Nowak et al. [26] pointed out that AP is a benign pathology with an excellent prognosis. Furthermore, the authors suggested not introducing the trigger food within six months of life due to the possible recurrence of bleeding. Furthermore, the authors stated that, generally, children with AP acquire tolerance between one and three years of life, and most tolerate food within one year of life. On the other hand, Arvola et al. [8], using clinical manifestations to guide the therapeutic approach, demonstrated that the elimination diet is effective only for a subset of patients whose signs and symptoms recur upon reintroduction of cow’s milk. Elizur et al. [6] demonstrated a low recurrence rate of clinical manifestations with the early reintroduction of cow’s milk (5–6 months) and suggested that the OFC should be performed shortly after clinical manifestations resolve to avoid the possibility of having false positives and, thus, the risk of overdiagnosis of AP and prolonged and unnecessary elimination diets. Based on these data, Miceli Sopo et al. [38] propose only three months of the elimination diet for milk and its subsequent reintroduction into the diet, as subsequently confirmed by Mennini et al. [2].

In conclusion, the times for the reintroduction of the offending food are not codified unanimously and vary considerably in the various studies. Usually, one year of age is used as a reference, but some authors [2,38], based on the study conducted by Elizur et al. [6], recommend trying again after three months of the diet. Also, in the study by Martin et al. [10], tolerance is established after an average of 50 days; however, in the previous study, the diagnosis of AP is overestimated because it considers even occult blood alone as symptomatology sufficient for the diagnosis of AP. In general, longer times for reintroduction could be reserved for the most severe cases or cases associated with IgE-mediated food allergy, for which it has been demonstrated that tolerance sets in later [34,35].

### 10.3. Setting

For children who have presented typical signs and symptoms, such as hematochezia, and who have been successfully treated with an extensively hydrolyzed or amino acid formula, the reintroduction of cow’s milk can take place in a protected environment or at home.

### 10.4. Mode of Introduction in a Hospital Setting

Various introduction schemes have been used [6,31]. In the work of Elizur et al. [6], the dose was increased starting from 1 mL of milk diluted 1:10 (2.7 mg of cow’s milk protein) up to 120 mL of pure milk (3.24 g of cow’s milk protein). The OFC was discontinued if any cutaneous, gastrointestinal, or systemic symptoms appeared. In the case of a negative OFC, the patient was observed for 3 h and discharged. Two weeks later, the patient was contacted by the health care provider to investigate his/her eating habits. In the work of Kaya et al. [31], the dose was increased every 15 min, from 0.1 to 1.0, 3.0, 10, 30, 50, and 100 mL of cow’s milk. If the OFC was negative, the patient continued to take the food at home, and the allergist contacted the patient to check for any late signs and symptoms and to investigate his/her eating habits.

### 10.5. Method of Introduction at Home

This varies depending on whether the infant is breastfed or formula fed. In the work of both Nowak-Węgrzyn et al. [26] and Miceli Sopo et al. [38], it was suggested that before proceeding with the OFC at home, the negativity of specific IgE for cow’s milk should be confirmed.

For breastfed infants: The mother inserts 30 mL of cow’s milk (or an equivalent dose for dairy products) into the diet and increases this by 30 mL each day until the diet is completely liberalized for cow’s milk [2,26,38].For formula-fed or no-longer-breastfed infants: 30 mL of cow’s milk is added to the amount of formula the infant is currently taking and increased by 30 mL every 2–3 days until the desired dose is reached [27]. This mode of gradual introduction is continued until an age-appropriate dose of cow’s milk is reached. The child is observed clinically to monitor for any recurrence of hematochezia, diarrhea, irritability, or other clinical manifestations. Any recurrence of the clinical manifestations generally occurs within 1–2 weeks of the introduction of the allergen. Reintroduction of cow’s milk is usually tolerated. If this fails, one option may be to introduce dairy-containing baked goods into the child’s diet before introducing pure cow’s milk [27].

### 10.6. Recurrence

If the hematochezia recurs during the introduction of the trigger food, the elimination diet must be resumed for a further 3 months, after which a new attempt to introduce the trigger food can be made [2,6].

## 11. Multidisciplinary Management of the AP

For the optimal management of the patient affected by AP, a multidisciplinary approach is indispensable, which involves several specialists, such as a pediatrician, allergist, gastroenterologist, and dietician [37]. In our opinion, the management of the patient with suspected AP can initially be the prerogative of the pediatrician, who, in the case of breastfeeding, can advise the mother to exclude cow’s milk from her diet or, in the case of bottle-feeding, can advise switching to an extensive hydrolysate or amino acid formula and refer the patient to the allergist. In some (mild) cases, the option of no special diet can be considered in agreement with the parents.

The allergist carries out an SPT on the trigger food excluded from the diet before the food’s reintroduction to decide the right setting for reintroduction (at home if allergy tests are negative or in a hospital setting if allergy tests are positive) and the timing, since a positive result puts the patient at risk of acquiring tolerance later. The allergist also confirms the diagnosis with the OFC and evaluates the exclusion of other allergens besides cow’s milk and the indication for an extensive hydrolysate or amino acid formula.

The allergist sends complex clinical cases to the gastroenterologist (e.g., those children who do not respond to the elimination diet for multiple allergens or an amino acid formula) to evaluate the need for further investigations, such as an endoscopic examination.

The dietician will play an important role in guiding the maternal diet in the event of an elimination diet for one or more of the indicated foods to ensure the correct intake of calories, calcium, and nutrients in the diet [27] (Table 3).

A psychologist can be involved to support mothers in case of anxiety, especially if multiple food diets are required.

## 12. Parental Involvement

It is necessary to involve the parents in the management of the child with AP, explaining to them the advantages and disadvantages of all options (Table 4).

It is important to consider that the state of anxiety of mothers of children with AP is very high, as demonstrated by the study by Sancakli et al. [68], in which the scores of mothers of patients with AP were higher (if they had to follow elimination diets for multiple foods) than those of mothers of healthy patients.

## 13. Prognosis

The prognosis of AP is good, as most infants can tolerate cow’s milk by one year of life [9,30,63,69]. According to a prospective observational study [10], the onset and diagnosis of AP is around one month of age, and tolerance is established on average after 50 days of an elimination diet. The prognosis is good even in cases where no diet is followed (15%).

Despite this, it has been shown that about 30–40% of patients acquire tolerance towards the culprit food after one year of life [26,31,32,33] and that another 5% develop tolerance after 3 years of life [32].

Senocak et al. [9] demonstrated that the patients who acquired tolerance later (>1 year of life) were those patients who presented with diarrhea, in association with hematochezia, as a clinical manifestation and who had been placed on a diet with an amino acid formula.

Buyuktiryaki et al. [34] demonstrated that the use of antibiotics in the first 6 months of life, the presence of colic, AP for multiple foods, and allergic sensitization to foods are risk factors for developing later tolerance.

Only 20% of breastfed infants with AP have been shown, in several studies, to have spontaneous resolution of clinical manifestations without changes in maternal diet [26,32,41]; this finding is not in line with the proposal to wait a month before starting the exclusion diet, with a view toward spontaneous resolution of the symptoms.

A prospective observational study on a cohort of healthy children demonstrated that children with AP have twice the risk of developing an IgE-mediated food allergy compared to children without AP and that milk is the food most associated with the development of an IgE-mediated allergy [70]. The exact conversion rate from AP to an IgE-mediated allergy to the same trigger food is not well known. One study [35] showed that 3.6% of patients affected by AP developed an IgE-mediated allergy to the same trigger food during follow-up and that these patients achieved statistically significant acquisition of tolerance toward milk later than patients who did not experience IgE-mediated conversion (19 months versus 11 months, *p* < 0.001). The same study found no risk factors predictive of conversion to an IgE-mediated allergy.

Other authors have shown that the risk of developing functional gastrointestinal disorders at the age of 4 was 4.39% (95% confidence interval 1.03–18.68) and was more frequent in patients who had more severe clinical manifestations of AP, with major hematochezia duration and younger age at onset of signs and symptoms [71].

Only one study [72] evaluated the evolution of AP into chronic inflammatory bowel disease and demonstrated that after 5–10 years, none of the 13 enrolled patients had developed chronic inflammatory bowel disease. To date, there are no other studies with longer follow-ups that evaluate this specific aspect.

## 14. Conclusions

AP is a benign condition that generally affects infants in the first few months of life, especially if they are exclusively breastfed. Cow’s milk is the trigger food most implicated in the pathogenesis of this condition. The diagnosis is clinical, and the treatment consists of the elimination diet of the trigger food for an appropriate period. Management must involve the pediatrician, allergy specialist, gastroenterologist, dietician, and sometimes the psychologist. Finally, parents must be actively involved in evaluating individual options. Most infants tolerate the trigger food in the first year of life. There is a growing need to better characterize this condition from a pathophysiological point of view to identify potential biomarkers able to help the clinician in making the diagnosis and able to predict the age of acquisition of tolerance to avoid unnecessary and prolonged elimination diets.

## Figures and Tables

**Figure 1 life-13-01824-f001:**
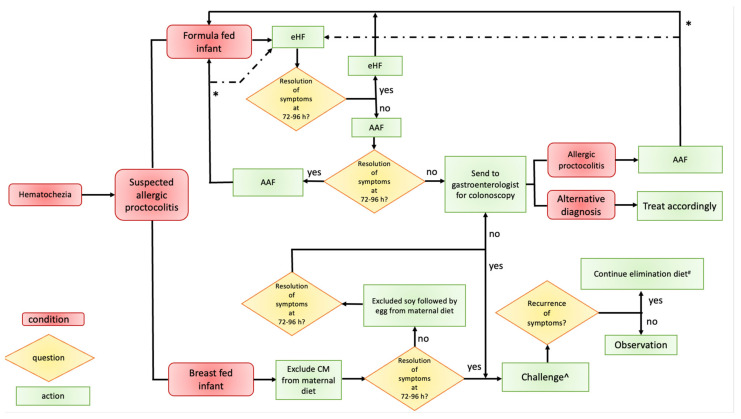
Proposed algorithm for diagnosis and treatment of allergic proctocolitis. Legend: AAF = amino acid formula; CM = cow’s milk; eHF = extensively hydrolyzed formula. # Under the supervision of a dietitian. ^ At home if skin prick test to trigger food is negative; in hospital setting if the skin prick test to trigger food is positive. * To change from an AAF directly to a cow’s milk infant formula, except in selected cases where one may consider changing AAF to eHF as an interim step before cow’s milk formula.

**Table 1 life-13-01824-t001:** Clinical and laboratory characteristics of allergic proctocolitis.

Age of onset of symptoms	First weeks to months of life (less than 6 months); it can occasionally occur in older children
Trigger foods	Most common: cow’s milk, soy Less common: egg, corn, wheat
Multiple food triggers	Occasional
Type of feeding at diagnosis	Breastfeeding (>60%)
Clinical manifestations	Presence of bright red blood with or without mucus in stool with or without diarrhea in otherwise healthy children Less common: flatulence, refusal to feed, abdominal colic
Atopic comorbidities	Eczema: 22–52% Atopic family history: 25–50%
Laboratory tests	Mild anemia Eosinophilia Increase in total IgE (occasional) Hypoalbuminemia (rare)
Stool exam	Eosinophils Visible or occult blood
Endoscopy/histology	Focal colitis, eosinophilic infiltrate, lymph node hyperplasia
Allergy test * (SPT and s-IgE)	Usually negative, positive in 10–35% of cases
Diagnosis	Clinical history and examination +/− OFC
Treatment	Avoidance of trigger food(s) (if breastfed, only consider exclusion of these foods in maternal diet)
Resolution of clinical symptoms with elimination diet	72–96 h
Natural history	Resolution in the first year of life

Legend: OFC = oral food challenge; * to be performed before reintroducing the food into the diet to decide on the right setting as a prognostic factor on the acquisition of tolerance.

**Table 2 life-13-01824-t002:** Dietary treatment in allergic proctocolitis.

Cow’s milk	In the breastfed infant: 1st choice: maternal elimination diet 2nd choice: eHF 3rd choice: AAF In the formula-fed infant: 1st choice: eHF 2nd choice: AAF if eHF fails
Soy	Elimination diet if there is no clinical response to cow’s milk elimination diet
Egg	Elimination diet in the case that there is no clinical response to the elimination of cow’s milk and soy

Legend: AAF = amino acid formula; eHF = extensively hydrolyzed formula.

**Table 3 life-13-01824-t003:** Healthcare professionals involved in the management of children with AP.

Pediatrician	- Manages the elimination diet in mild cases - Refers to allergist specialist
Allergologist	- Performs allergy tests to decide reintroduction settings and to assess prognosis - Confirm diagnosis with OFC - Considers prescribing eHF or AAF
Gastroenterologist	- Handles complex clinical cases - Performs endoscopic examination if necessary
Dietitian	- Supports patients with prolonged and/or multi-food diets
Psychologist	- Possible support to mothers in case of anxiety, especially if exclusion diets involve multiple foods

Legend: AAF = amino acid formula; eHF = extensively hydrolyzed formula; OFC = oral food challenge.

**Table 4 life-13-01824-t004:** Advantages and disadvantages of different therapeutic options in the management of patients with AP.

Therapeutic Options	Advantages	Disadvantages
Empiric diet	- Simple and effective - Can be managed by the pediatrician	- Risk of overdiagnosis - No definitive diagnosis - Often longer than necessary
Elimination diet after diagnostic OFC	- Excludes false positives - Diagnosis of certainty	- Recurrence of bleeding after a period of well-being
“Watchful waiting” (no change to diet)	- Avoids the elimination diet - More economical (avoids any eHF or AAF)	- Parental anxiety - Only 20% of patients overcome hematochezia without elimination diet - Possible occurrence of long-term anemia

Legend: AAF = amino acid formula; eHF = extensively hydrolyzed formula; OFC = oral food challenge.

## Data Availability

Not applicable.

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
