# Peer review of "Allergic Proctocolitis: Literature Review and Proposal of a Diagnostic–Therapeutic Algorithm"

_life, 2023, doi:10.3390/life13091824_

Round 1
Reviewer 1 Report
The review article Allergic Proctocolitis: literature review and Proposal of a Diagnostic-therapeutic Algorithm is clearly written and well organized—however some suggestions to the authors.
1. Page no 9 Arvola et al study shows no difference between the cow's milk elimination group and the unchanged diet group in duration and severity of bleeding and hematochezia needs more explanation.
2. It will be more helpful if the authors explain more about the mechanistic role of TGF-beta, TNF-alfa, and Interleukins in Allergic proctocolitis.
Grammatical mistakes and minor editing is required.
Author Response
- Page no 9 Arvola et al study shows no difference between the cow's milk elimination group and the unchanged diet group in duration and severity of bleeding and hematochezia needs more explanation.
Thank you for your observation. We will include a more extensive explanation.
- It will be more helpful if the authors explain more about the mechanistic role of TGF-beta, TNF-alfa, and Interleukins in Allergic proctocolitis.
Thank you for your observation. We will include more details regarding the role of interleukins in Allergic proctocolitis.
Reviewer 2 Report
A very well designed and prepared work. I am very glad that you sent me this study for consideration. I think this study will shed light on many physicians evaluating intestinal bleeding patients. However, in the case of rectal bleeding in children, there are two very important diseases that should be considered in the differential diagnosis.
1-Henoch-Schönlein Purpura in Children is an uncommon pathology but can cause life-threatening bleeding and its treatment is specific. It should be considered in the differential diagnosis and the pathologist who examines the biopsy taken should be warned because it is a vasculitis. Leung AKC, Barankin B, Leong KF. Henoch-Schönlein Purpura in Children: An Updated Review. Curr Pediatr Rev. 2020;16(4):265-276. doi: 10.2174/1573396316666200508104708. PMID: 32384035.
2-Although it can be seen very rarely in developed countries such as Italy, intestinal parasitosis is a very important cause of intestinal bleeding in the Middle East and Asia. It must necessarily be included in the differential diagnosis. Sangkhathat S, Patrapinyokul S, Wudhisuthimethawee P, Chedphaopan J, Mitamun W. Massive gastrointestinal bleeding in infants with ascariasis. J Pediatr Surg. 2003 Nov;38(11):1696-8. doi: 10.1016/s0022-3468(03)00584-0. PMID: 14614730.

Author Response
A very well designed and prepared work. I am very glad that you sent me this study for consideration. I think this study will shed light on many physicians evaluating intestinal bleeding patients. However, in the case of rectal bleeding in children, there are two very important diseases that should be considered in the differential diagnosis.
1-Henoch-Schönlein Purpura in Children is an uncommon pathology but can cause life-threatening bleeding and its treatment is specific. It should be considered in the differential diagnosis and the pathologist who examines the biopsy taken should be warned because it is a vasculitis. Leung AKC, Barankin B, Leong KF. Henoch-Schönlein Purpura in Children: An Updated Review. Curr Pediatr Rev. 2020;16(4):265-276. doi: 10.2174/1573396316666200508104708. PMID: 32384035.
2-Although it can be seen very rarely in developed countries such as Italy, intestinal parasitosis is a very important cause of intestinal bleeding in the Middle East and Asia. It must necessarily be included in the differential diagnosis. Sangkhathat S, Patrapinyokul S, Wudhisuthimethawee P, Chedphaopan J, Mitamun W. Massive gastrointestinal bleeding in infants with ascariasis. J Pediatr Surg. 2003 Nov;38(11):1696-8. doi: 10.1016/s0022-3468(03)00584-0. PMID: 14614730.
Thank you for your observation. We will include in the article the two diseases that should be considered in the differential diagnosis